# Determining $\alpha_s$ from hadronic $\tau$ decay: the pitfalls of truncating the OPE

D. Boito[1], M. Golterman[2], K. Maltman[3,4], S. Peris[5,*]

**1** Instituto de Física de São Carlos, Universidade de São Paulo, Brazil
**2** Department of Physics and Astronomy, San Francisco State University, USA
**3** Department of Mathematics and Statistics, York University, Canada
**4** CSSM,University of Adelaide, Adelaide, Australia
**5** Department of Physics and IFAE-BIST, Universitat Autònoma de Barcelona, Spain
*peris@ifae.es

November 6, 2018

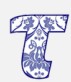 *Proceedings for the 15th International Workshop on Tau Lepton Physics, Amsterdam, The Netherlands, 24-28 September 2018*
scipost.org/SciPostPhysProc.Tau2018

## Abstract

We discuss sum-rule determinations of $\alpha_s$ from non-strange hadronic $\tau$-decay data. We investigate, in particular, the reliability of the assumptions underlying the "truncated OPE strategy," which specifies a certain treatment of non-perturbative contributions, and which was employed in Refs. [1–3]. Here, we test this strategy by applying the strategy to the $R$-ratio obtained from $e^+e^-$ data, which extend beyond the $\tau$ mass, and, based on the outcome of these tests, we demonstrate the failure of this strategy. We then present a brief overview of new results on the form of duality-violating non-perturbative contributions, which are conspicuously present in the experimentally determined spectral functions. As we show, with the current precision claimed for the extraction of $\alpha_s$, including a representation of duality violations is unavoidable if one wishes to avoid uncontrolled theoretical errors.

# 1 Introduction

As is well known, the determination of $\alpha_s$ from finite-energy sum-rule (FESR) analyses of hadronic $\tau$-decay data provides one of the most precise determinations of $\alpha_s$. Because of its low scale, this determination, moreover, plays an important role in testing the evolution of the strong coupling predicted by QCD. In this paper we pull back the curtain on, and subject to further scrutiny, certain issues and subtleties connected with the treatment of non-perturbative effects; issues which the precision now claimed for these determinations makes it important to understand in more quantitative detail.

In what follows, we first demonstrate that certain highly non-trivial assumptions made in treating non-perturbative contributions in common implementations of the FESR analysis framework can be tested (and shown to fail) via analogous analyses of electromagnetic (EM) hadroproduction cross-sections, which, unlike hadronic $\tau$-decay distributions, are not kinematically restricted to hadronic invariant-squared-masses $s \leq m_\tau^2$. These observations imply that $\tau$-decay analyses cannot avoid employing weighted spectral integrals with variable upper endpoints, $s_0 \leq m_\tau^2$. Given that significant duality violations (DVs) are clearly observed in the experimental differential non-strange hadronic $\tau$-decay distributions, this necessitates providing estimates for the size of residual DV effects, which in turn necessitates the use of models for the DV components of the hadronic spectral functions. Recent progress in determining the form of these components expected in QCD, relevant to carrying out such analyses, is then also reviewed.

In the Standard Model, defining

$$R_{ud;V/A} \equiv \frac{\Gamma[\tau \to \nu_\tau\, \text{hadrons}_{ud;V/A}\,(\gamma)]}{\Gamma[\tau^- \to \nu_\tau e^- \bar{\nu}_e(\gamma)]} \ , \tag{1}$$

one has [4]

$$\frac{dR_{ud;V/A}}{ds} = \frac{12\pi^2\, |V_{ud}|^2 S_{EW}}{m_\tau^2} \left[ w_\tau\left(y_\tau\right) \rho_{ud;V/A}^{(0+1)}(s) - w_L\left(y_\tau\right) \rho_{ud;V/A}^{(0)}(s) \right] \ , \tag{2}$$

where $y_\tau = s/m_\tau^2$, $w_\tau(y) = (1-y)^2(1+2y)$, $w_L(y) = 2y(1-y)^2$, $V_{ud}$ is the $ud$ element of the CKM matrix, $S_{EW}$ is a known short-distance electroweak correction [5], and $\rho_{ud;V/A}^{(J)}(s)$ are the spectral functions of the $J = 0, 1$ hadronic vacuum polarizations (HVPs), $\Pi_{ud;V/A}^{(J)}$, of the flavor $ud$, vector $(V)$ and axial-vector $(A)$ current-current two-point functions. The continuum parts of $\rho_{ud;V/A}^{(0)}(s)$ are suppressed by factors of $(m_d \mp m_u)^2$, and hence numerically negligible, leaving the well-determined pion-pole contribution to $\rho_{ud;A}^{(0)}$ as the only numerically relevant $J = 0$ contribution. The $J = 0 + 1$ sums $\rho_{ud;V/A}^{(0+1)}(s)$ are thus directly determinable from the experimental $dR_{ud;V/A}/ds$ distributions.

The spectral function combinations $\rho_{ud;V/A}^{(0+1)}(s)$ and $s\, \rho_{ud;V/A}^{(0)}(s)$ correspond to HVP combinations, $\Pi_{ud;V/A}^{(0+1)}(s)$ and $s\Pi_{ud;V/A}^{(0)}(s)$, which are free of kinematic singularities. For any $s_0 \leq m_\tau^2$, and any weight $w$ analytic inside and on $|s| = s_0$, Cauchy's theorem, applied to the contour in Fig. 1 then ensures the validity of the FESR relation [6]

$$\int_0^{s_0} \frac{ds}{s_0}\, w(s/s_0)\, \rho_{ud;V/A}^{(0+1)}(s) = \frac{-1}{2\pi i} \oint_{|s|=s_0} \frac{ds}{s_0}\, w(s/s_0)\, \Pi_{ud;V/A}^{(0+1)}(s) \ . \tag{3}$$

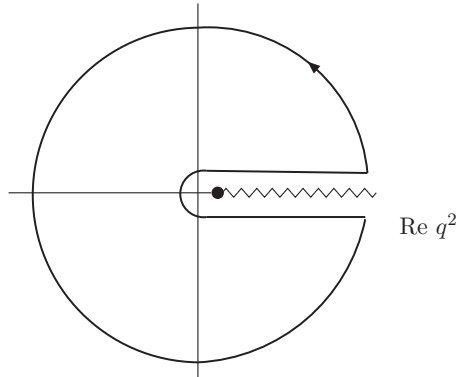

Figure 1: *Contour used in the derivation of Eq. (3). The cut shown on the positive real $s = q^2 = -Q^2$ axis starts at $s = 4m_\pi^2$ for $\Pi_{ud;V}^{(0+1)}$ and $s = 9m_\pi^2$ for $\Pi_{ud;A}^{(0+1)}$. $\Pi_{ud;A}^{(0+1)}$, of course, also has a pole at $s = m_\pi^2$.*

The basic idea of the $\tau$-based determination of $\alpha_s$ is to employ experimental results for $dR_{ud;V/A}/ds$ on the LHS of Eq. 3 and, for sufficiently large $s_0$, the Operator Product Expansion (OPE) representation of $\Pi_{ud;V/A}^{(0+1)}(s)$ on the RHS. The OPE, of course, represents only an approximation to $\Pi_{ud;V/A}^{(0+1)}$. In general, in addition to perturbative (dimension $D = 0$ OPE) and higher dimension non-perturbative OPE condensate contributions,

$$\left[\Pi_{ud;V/A}(s)\right]_{OPE}^{NP} = \sum_{D=4,6,8,\cdots} \frac{C_D^{V/A}}{Q^D} , \tag{4}$$

with $Q^2 = -s$, and the $C_D^{V/A}$ effective condensates of dimension $D$, non-OPE, DV contributions, $\left[\Pi_{ud;V/A}^{(0+1)}(s)\right]_{DV}$, defined by

$$\Pi_{ud;V/A}^{(0+1)}(s) \equiv \left[\Pi_{ud;V/A}^{(0+1)}(s)\right]_{OPE} + \Pi_{ud;V/A}^{DV}(s) , \tag{5}$$

are needed to provide a full representation of $\Pi_{ud;V/A}^{(0+1)}(s)$.

If $m_\tau^2$ were sufficiently large that, relative to perturbative contributions, all non-perturbative contributions (both DV and OPE) were negligible on the circle $|s| = m_\tau^2$, the inclusive experimental non-strange hadronic $\tau$ decay width would provide an immediate determination of $\alpha_s$. Unfortunately, this is not the case, at the level of precision desired (and claimed) in current $\tau$-based analyses.

Two key qualitative points should be emphasized regarding non-perturbative contributions to the RHS of Eq. (3). First, since the cut in $\Pi_{ud;V/A}^{(0+1)}$ extends to $s = \infty$ ($z = 1/Q^2 = 0$), the OPE (an expansion in $z$ about $z = 0$) cannot be convergent. Second, DV contributions to $\Pi_{ud;V/A}^{(0+1)}(s)$, which are exponentially suppressed for large spacelike $Q^2 = -s$, are expected to develop an additional oscillatory behavior on the Minkowski axis [7,8]. Such oscillations

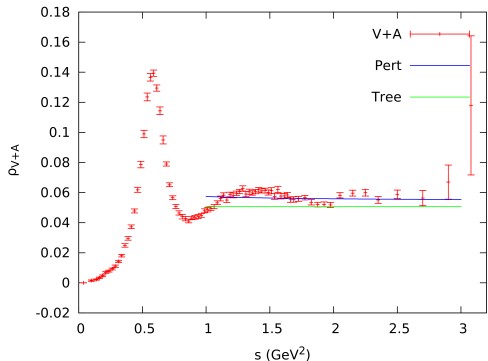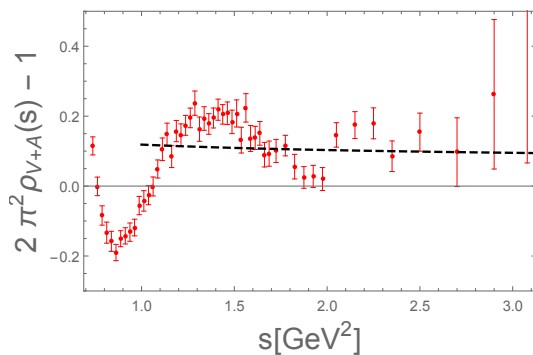

Figure 2: *Left panel: the ud $V + A$ spectral function, as shown in Ref. [1]. Lower panel: the same (in the normalization of [9]) but now with the $\alpha_s$-independent parton-model contribution subtracted.*

are clearly seen in $\rho_{ud;V/A}^{(0+1)(s)} = \frac{1}{\pi} \operatorname{Im} \Pi_{ud;V/A}^{(0+1)}(s)$ but their properties are not captured by the OPE. They reflect the incipient presence of resonances as the energy is lowered from the parton-model regime.

The fact that the OPE is not convergent means that it is not true that higher dimension OPE contributions to the RHS of Eq. (3) scale simply as $\Lambda_{QCD}^{D}/s_0^{D/2}$ and hence form a rapidly converging series in $D$ for $s_0 \simeq m_\tau^2$. Assuming, on such "dimensional" grounds, that integrated higher-$D$ OPE contributions in principle present for a given weight $w$ can be neglected requires experimental justification if one wishes to avoid incurring unquantifiable systematic errors.

The fact that DV effects are not, in general, negligible in hadronic $\tau$ decays is evidenced by the size of the observed DV oscillations in the $V$, $A$ and $V + A$ spectral functions. It is often argued [1] that DV oscillations are "small" for the $V + A$ combination on the basis of plots showing the size of such oscillations on the scale of the full $V + A$ spectral function, $\rho_{ud;V+A}^{(0+1)}(s)$, as in the left panel of Fig. 1. Such a plot is, however, highly misleading, since $\rho_{ud;V+A}^{(0+1)}(s)$ contains a large parton-model contribution completely independent of $\alpha_s$, *i.e.*, of all QCD dynamics. The FESR determination of $\alpha_s$ is driven entirely by the dynamical, $\alpha_s$-dependent part of the perturbative contribution to the weighted spectral integrals, and the relevant measure of the relative size of perturbative and DV contributions to the spectral functions entering those integrals, from the point of view of a determination of $\alpha_s$, is the size of the DV oscillations relative to the $\alpha_s$-*dependent part* of the perturbative representation of $\rho_{ud;V+A}^{(0+1)}(s)$. The right panel of Fig. 1 shows this more relevant comparison. One immediately sees, for example, that the non-parton-model part of $\rho_{ud;V+A}^{(0+1)}(s)$ is $\simeq 0$ for $s \simeq 2$ GeV$^2$, indicating that DV and $\alpha_s$-dependent perturbative contributions are, in fact, equal in magnitude in this region, essentially cancelling each other out. This is also true in the vicinity of the next DV peak, where, however, the two contributions combine constructively, as expected given the oscillatory nature of DVs. While it *is* true that DV oscillations are smaller for the $V + A$ combination than for the individual $V$ and $A$ spectral functions, this rather obviously does not mean that the $V + A$ oscillations are small in an absolute sense.

DV contributions, though important in hadronic spectral functions, certainly for the range of $s$ accessible in $\tau$ decays, may be suppressed relative to perturbative contributions when one considers the integrated quantities appearing on the RHSs of Eq. (3). From the arguments of Ref. [10], DV contributions on $|s| = s_0$ at intermediate $s_0$ are expected to be localized to the vicinity of the timelike axis. Given the asymptotic nature of the OPE, and the oscillatory behavior of the DVs, the parametrization [11–13]

$$\frac{1}{\pi} \mathrm{Im}\Pi^{DV}_{ud;V/A}(s) = e^{-\delta_{V/A} - \gamma_{V/A} s} \sin\left(\alpha_{V/A} + \beta_{V/A} s\right) \ , \tag{6}$$

for $s$ large enough, represents a very natural choice.[1] In fact, this expression has recently been confirmed [8] under the mild assumption of an asymptotic Regge behavior for the meson spectrum. We discuss this in more detail in section 3 below.

The contribution from $\Pi^{DV}_{ud;V/A}(s)$ in Eq. (5) to the FESR (3) can be shown to take the form [11, 14]

$$\frac{-1}{2\pi i} \oint_{|s|=s_0} \frac{ds}{s_0} \, w(s/s_0) \, \Pi^{DV}_{ud;V/A}(s) = -\int_{s_0}^{\infty} \frac{ds}{s_0} \, w(s/s_0) \, \frac{1}{\pi} \mathrm{Im}\, \Pi^{DV}_{ud;V/A}(s) \ , \tag{7}$$

and, using the parametrization (6), this form of the DV contributions on the RHS of (7) will be very useful in the following discussion.

Because of the exponential suppression at large $s$ in Eq. (6), and localization of DV contributions to the vicinity of the timelike axis at intermediate and large $s$, the use of "pinched weights" $w$ (those with a zero at $s = s_0$) in Eq. (3) is thus expected to yield RHSs in the FESRs (3) in which residual integrated DVs play a reduced role relative to integrated OPE contributions, with the level of suppression typically increasing with the degree of pinching (the order of the zero at $s = s_0$). This general expectation is confirmed empirically [15], and can also be seen on average, over a range of $s_0$, when one integrates explicitly the asymptotic DV form (6) [14]. While increased pinching increasingly suppresses DVs on average, given the size of DV contributions to the spectral functions, and the precision claimed for current versions of the determination of $\alpha_s$, it remains an open question how large this suppression is for the doubly and triply pinched weights employed in typical determinations of $\alpha_s$.

The analyses of Refs. [1–3], all implicitly assume the scale $s_0 = m_\tau^2$ is high enough that integrated DVs can be neglected for the doubly and triply pinched weights entering those analyses. Ref. [16] also assumes integrated DVs can be neglected for the doubly pinched weights it employs, testing this assumption for self-consistency by studying the $s_0$ dependence of the resulting fits. In contrast, the analysis of Ref. [17, 18] employs the model for DV contributions to the $V$ and $A$ spectral functions of Eq. (6), and finds a systematic downward shift in the $\alpha_s$ obtained when this representation of DV effects is included.

Of particular relevance to the results of Refs. [1,16], where attempts are made to investigate the self-consistency of the assumed neglect of DVs, are the results of Ref. [9], where the tests employed in Ref. [1] are applied to a model based on mock data which accurately matches the experimental $ud\ V + A$ spectral function and which has, by construction, a lower input value of $\alpha_s$ as well as numerically relevant DV contributions at higher $s$. It is found that what were hoped to be self-consistency tests in Ref. [1], applied to this model, are unable to identify the presence of the model DV contributions and the lower input $\alpha_s$ value, establishing

---

[1]Recall how renormalons give rise to a $e^{-b/\alpha_s}$ behavior from the asymptotic nature of perturbation theory. Here the expansion parameter is $1/s$ rather than $\alpha_s$ and $\alpha_s$ is thus parametrically replaced by $1/s$.

that these tests are, in general, insufficient to establish the absence of numerically significant DVs. A comparison of the results of Refs. [16] and [17] suggests this same caveat is relevant to assessing the results of the $s_0$-dependence tests employed in Ref. [16].

The failure of nominal self-consistency tests of Ref. [1] when applied to the model described above still leaves open the logical possibility that DV contributions to the actual spectral functions in the region $s > m_\tau^2$ might be smaller than those in the model, leading to smaller integrated DV contributions in the real world than in the model. This possibility can be investigated, at least for the $I = 1$ $V$ component, using $e^+e^-$ hadroproduction cross-section data [19]. The reason is that the $I = 1$ component of the EM current is related by CVC to the charged $I = 1$ $V$ current acting in $\tau$ decays. The predictions of the $\tau$-based model of Ref. [9] for the $I = 1$ spectral function in the region $s_0 \geq m_\tau^2$, where it cannot be measured in $\tau$ decays, can then be tested against the $I = 1$ component of the EM spectral function obtained from a $G$-parity based isovector/isoscalar separation of the $I = 0$ and $I = 1$ contributions to the EM spectral function. This separation was carried out in the region up to $s = 4$ GeV$^2$ in Ref. [20]. The result of the comparison to the prediction shown in Fig. 5 of Ref. [20] shows good agreement: the DV oscillations predicted by the $\tau$-decay based model are, indeed, seen in the $e^+e^-$ data.

One can also use the electroproduction $R(s)$ data to test the "truncated OPE strategy" which is the foundation for the analysis of Refs. [1,2]. The problem with the truncation of the OPE arises as follows. The spectral integral for the $s_0 = m_\tau^2$ version of the FESR involving the kinematic weight $w_\tau(y)$, with $y \equiv s/s_0$, can directly be determined from the inclusive branching fraction for hadronic $\tau$ decays. This result, however, is insufficient to allow one to determine $\alpha_s$ since $w_\tau$ has degree 3, and the theorem of residues then implies that the right-hand (theory) side of the $w_\tau$ FESR involves OPE contributions up to $D = 8$. While the $D = 4$ contribution is strongly suppressed by the absence of a term linear in $y$ in $w_\tau(y)$, three OPE parameters, $\alpha_s$, and the $D = 6$ and 8 effective condensates, $C_6$ and $C_8$, are still required to fix the theory side. Since $C_6$ and $C_8$ are not known from external sources, the inclusive non-strange branching fraction itself cannot provide a determination of $\alpha_s$.

A strategy employed to try to get around this problem [1–3] is to consider additional $s_0 = m_\tau^2$ FESRs involving new, higher-degree weights, with at least the level of pinching of $w_\tau$. The goal is to use the additional weighted spectral integrals as inputs to an extended multi-weight analysis in which non-perturbative condensates like $C_6$ and $C_8$ are also fit.

This strategy, however, has a fundamental shortcoming. If one considers, for example, using one additional FESR involving a polynomial weight of degree 4, that FESR now receives a contribution from a new effective OPE $D = 10$ condensate, $C_{10}$. Adding a weight with degree 5 similarly brings into play a contribution proportional to another new $D = 12$ condensate, $C_{12}$, *etc.* As long as one aims to suppress as much as possible residual integrated DV effects by considering spectral integrals with $s_0 = m_\tau^2$ with at least doubly pinched weights only, one has, at every stage, more OPE parameters to fit than weighted spectral integrals to use in fitting them.

For this strategy to work in practice, one thus needs to make the strong additional assumption that, for a set of weights whose maximum degree is $N$, and which, therefore, requires knowledge of OPE condensates up to dimension $D = 2N + 2$, the OPE can be truncated at a dimension smaller than $2N + 2$ sufficiently low to leave the number of OPE fit parameters less than the number of spectral integrals to be used in fitting them. Though this truncation leads to a proper fit in the statistical sense, it is really only justified if the asymptotic OPE series behaves, at the scales of the analysis, as if it were convergent. This approach to determining

$\alpha_s$ from hadronic $\tau$-decay data, which we refer to as the "truncated OPE strategy," has been employed, for example, in Refs. [1–3].

The truncated OPE strategy is therefore predicated on the assumptions that $s_0 = m_\tau^2$ is large enough that (i) integrated DV contributions can be neglected for FESRs involving doubly and triply pinched weights and (ii) the OPE, though asymptotic at best, behaves as if it were rapidly convergent for dimensions up to $2N + 2$, where $N$ is the degree of the highest-degree weight entering the analysis in question.

It is important to stress that integrated DV contributions are expected to be exponentially damped with increasing $s_0$, and that integrated higher-dimension $D = 2k$ OPE contributions scale as $1/s_0^k$ and hence also decrease, relative to the leading $D = 0$ perturbative contributions, with increasing $s_0$.[2] It therefore follows that, if the assumptions of the truncated OPE strategy were valid for $s_0 = m_\tau^2$, they would be even more so for higher $s_0$. The kinematic restriction $s_0 \leq m_\tau^2$ unfortunately prevents this prediction from being tested using $\tau$-decay data but, fortunately, the $R$-ratio data obtained from $e^+e^- \to$ hadrons allow for such tests.

Analogous EM FESRs, employing results for $R(s)$ obtained in Ref. [19], thus allow us to investigate the reliability of the assumptions underlying the truncated OPE strategy by applying the same fits to the correspondingly weighted $s_0 = m_\tau^2$ versions of the EM spectral integrals, and then testing whether the resulting OPE fit results provide a good representation of the actual EM spectral integrals for $s_0 > m_\tau^2$. This investigation will be the subject of the next section.

## 2    $e^+e^-$-based tests of the truncated-OPE FESR strategy

In this section we focus our investigation of the assumptions underlying the truncated OPE strategy on two of the sets of weights employed in the nominal self-consistency studies of Ref. [1], namely the conventional "$(kl)$ spectral weight" set,

$$w_{kl}(y) = y^l (1 - y)^k \, w_\tau(y) \; , \tag{8}$$

with $(kl) = (00)$, $(10)$, $(11)$, $(12)$ and $(13)$, and the set of so-called "optimal weights",

$$w^{(2n)}(y) = 1 - (n + 2)y^{n+1} + (n + 1)y^{n+2} \; , \tag{9}$$

with $n = 1, \cdots, 5$.

The (00) spectral weight is doubly pinched and the remainder of the $(kl)$ spectral weights triply pinched, while the optimal weights are all doubly pinched. Since both the $(kl)$ spectral weight and optimal weight sets include weights up to degree 7, the corresponding sets of FESRs involve, in principle, OPE contributions, unsuppressed by additional factors of $\alpha_s$, up to $D = 16$. In order to leave one more $s_0 = m_\tau^2$ spectral integral than OPE parameter in the corresponding multi-weight fits, spectral-weight analyses fit $\alpha_s$, $C_4$, $C_6$ and $C_8$ and assume contributions proportional to $C_{10}$, $C_{12}$, $C_{14}$ and $C_{16}$ can be neglected. The absence of a term linear in $y$ in the weights $w^{(2n)}(y)$ means that contributions proportional to $C_4$ are strongly suppressed. The five $s_0 = m_\tau^2$ optimal-weight-set spectral integrals are then used to fit the four OPE parameters, $\alpha_s$, $C_6$, $C_8$ and $C_{10}$, with contributions proportional to $C_{12}$, $C_{14}$ and $C_{16}$ assumed negligible.

---

[2] Although the asymptotic nature of the OPE leads to the expectation of a rapid increase of the condensate contribution with its dimension $D$.

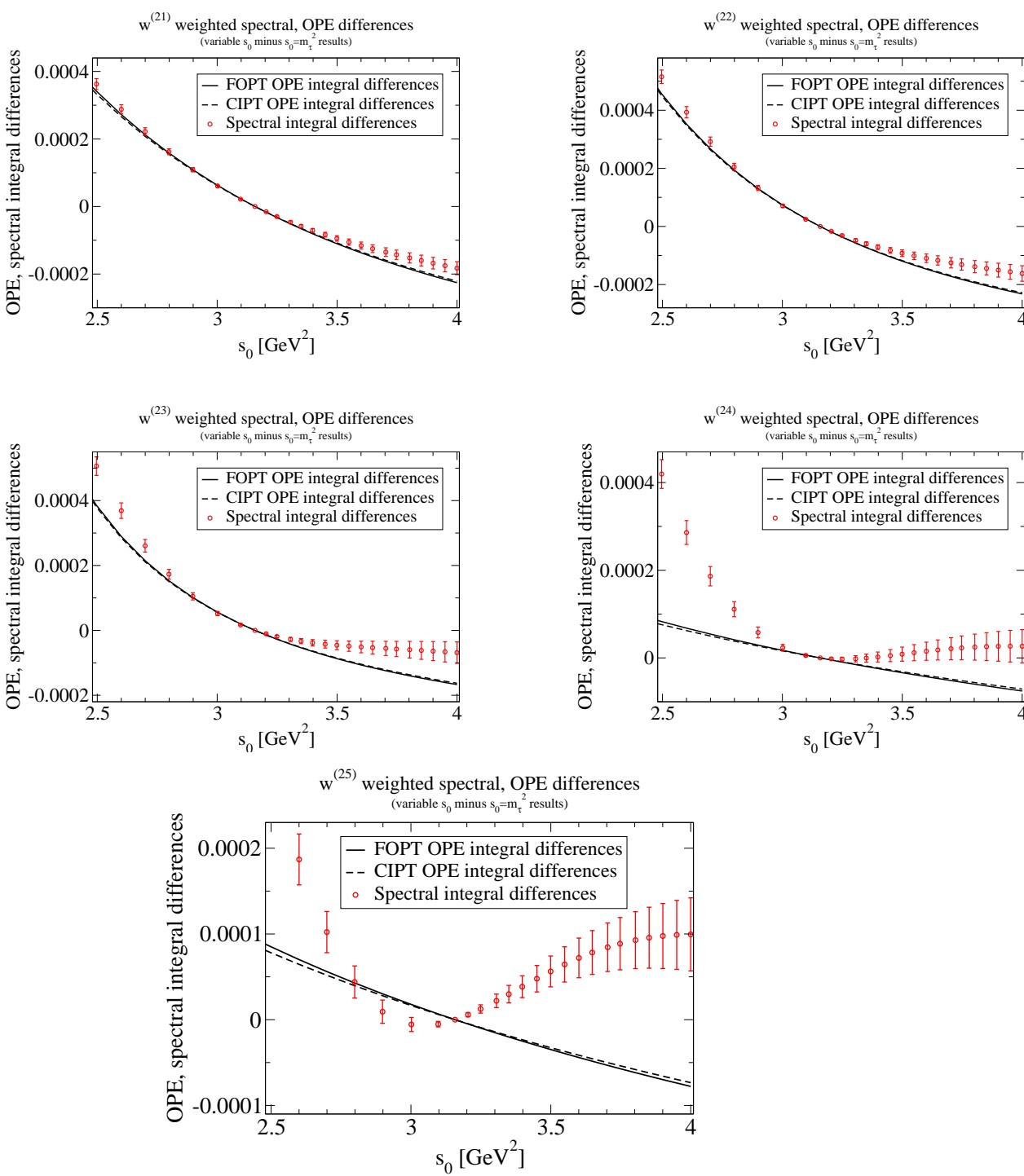

Figure 3: *EM FESR tests of the optimal weight version of the truncated OPE strategy. Comparisons of differences between general $s_0$ and $s_0 = m_\tau^2$ versions of the OPE and spectral integrals, with OPE results corresponding to OPE parameter values obtained from the optimal-weight implementation of the truncated OPE strategy using $s_0 = m_\tau^2$ only in the fits.* 8*Top left: $w^{(21)}$; top right: $w^{(22)}$; middle left: $w^{(23)}$; middle right: $w^{(24)}$; bottom: $w^{(25)}$.*

We consider spectral-weight and optimal-weight FESRs, in which the $ud$ $V$ or $A$ spectral functions and HVPs appearing in Eq. (3) are replaced by the corresponding spectral function, $\rho_{EM}(s)$, and HVP, $\Pi_{EM}(s)$, of the three-flavor EM current. The spectral function, $\rho_{EM}(s)$, is related to the well-known $R(s)$ ratio by

$$\rho_{EM}(s) = \frac{1}{12\pi^2} R(s) \ . \tag{10}$$

We employ the results and covariances for $R(s)$ provided by the authors of Ref. [19]. Full details of our own implementation of such EM FESRs may be found in Ref. [20].

We stress that (i) in the isospin limit, CVC implies that the $I = 1$ part of $\Pi_{EM}(s)$, $\Pi_{EM}^{I=1}$ is equal to $\frac{1}{2} \Pi_{ud;V}^{(0+1)}(s)$, where the $1/2$ is a trivial Clebsch-Gordon factor, and (ii) the $I = 0$ part of $\Pi_{EM}(s)$ is, up to a factor of $1/3$, the $SU(3)_F$ hypercharge partner of the $I = 1$ component. Higher dimension $I = 0$ EM OPE condensate contributions to $\Pi_{EM}(s)$ should thus be $\sim 1/3$ of the corresponding $I = 1$ EM OPE condensate contributions, up $SU(3)_F$ breaking effects. If $I = 1$ condensates of a given dimension yielded contributions which are negligible, relative to perturbative contributions, for $s_0 \geq m_\tau^2$, this should be equally true of the corresponding $I = 0$ contributions. It follows that, if the truncated-OPE-strategy assumptions were reliable at $s_0 = m_\tau^2$ for $\tau$-decay-based FESRs, they should be similarly reliable at $s_0 = m_\tau^2$ for the corresponding EM FESRs, and they should then be even more reliable for $s_0 > m_\tau^2$, though this expectation can only be tested in the EM case.

Of course, the EM case allows us to consider only the $V$ channel, whereas Ref. [1] considers $V + A$ to be the optimal choice for the truncated OPE strategy. We note, however, that (i) there is not a vast difference between the amplitude of the DV oscillations in the $V$ and $V + A$ channels, relative to the parton model, and (ii), that, in particular for the optimal weights, the results for $\alpha_s$ obtained in Ref. [1] on the basis of the truncated OPE strategy are in excellent agreement between fits to the $V$ and $V + A$ channels, while the corresponding agreement for the spectral weights is also very good.

We test the truncated OPE strategy by first performing truncated-OPE-strategy fits to the $s_0 = m_\tau^2$ versions of either the five $w_{kl}$-weighted EM spectral integrals or the five $w^{(2n)}$-weighted EM integrals, and then comparing the weighted EM spectral integrals and OPE integrals obtained using the resulting fitted OPE parameters at $s_0 > m_\tau^2$.

Very strong correlations exist between weighted spectral integrals for different $s_0$, as well as between weighted OPE integrals for different $s_0$. In order to take these correlations into account in assessing, visually, how successful the resulting $s_0 > m_\tau^2$ OPE integrals are in predicting the actual values of the corresponding EM spectral integrals, it is useful to plot not the spectral and OPE integrals themselves, but rather the difference between their values at general $s_0$ and $s_0 = m_\tau^2$. Both the OPE and spectral integral differences are thus zero, by definition, at $s_0 = m_\tau^2$. The errors on the spectral integral differences are straightforwardly obtainable from the covariance matrix of the $R(s)$ data provided by the authors of Ref. [19].

The results of this test are shown in Fig. 1, for the optimal-weight set of Ref. [1]. The OPE integral differences produced using the truncated-OPE-strategy fit assumptions obviously provide an, in general, very poor representation of the corresponding spectral integral differences in the region above $s_0 = m_\tau^2$. For the sake of brevity, the OPE-spectral integral matches of the analogous spectral-weight test, which are similarly bad above $s_0 = m_\tau^2$, are not shown here.

From these results it is clear that the assumptions underlying the truncated OPE strategy are, simply, not valid, and thus that results obtained from the truncated OPE strategy are

unreliable. Since the weights involved in these tests are doubly and/or triply pinched, and hence expected to have suppressed integrated DV contributions, especially above $s_0 = m_\tau^2$, the poor OPE-spectral integral matches imply a breakdown of the assumption that the OPE can be truncated as it would were the OPE a rapidly converging expansion up to at least $D = 16$.

The consequences of this observation for $\tau$-based analyses are (i) that the truncations in dimension of the OPE employed in the truncated OPE strategy are completely unsafe and (ii) that, in order to have fewer OPE parameters than spectral integrals required to fit them, one must consider also spectral integrals involving whatever set of weights one is employing at $s_0$ different from $m_\tau^2$, which, for analyses of $\tau$-decay data, means $s_0 < m_\tau^2$. Since quite sizeable DV oscillations about perturbation theory are observed in the spectral functions in this region, even when one considers the $ud\ V + A$ sum, it becomes important to use some representation of DV contributions to estimate the impact of possible residual DV effects, even in FESRs involving doubly and triply pinched weights.

## 3 Duality Violations and Hyperasymptotics: The Regge Connection

Although one expects DVs to behave as in Eq. (6) for large $s$ on general grounds, it would be nice to derive an expression such as Eq. (6) from QCD. Regrettably, this is still not possible from first principles but, recently, in Ref. [8], progress has been made under two plausible assumptions: (i) that the radial spectrum of QCD shows a leading Regge behavior in the vector channel for asymptotically large excitation number $n$, *i.e.*,

$$
\begin{aligned}
M^2(n) &= \Lambda_{QCD}^2\, n + b \log n + c + \mathcal{O}\left(\frac{1}{n}, \frac{1}{\log n}\right)\,, \\
\frac{F(n)}{F_0} &= 1 + \mathcal{O}\left(\frac{1}{n}, \frac{1}{\log n}\right)\,, \qquad (n \gg 1)\,,
\end{aligned}
\tag{11}
$$

where $M(n)$ is the spectrum of masses and $F(n)$ are the corresponding decay constants appearing in the vector two-point function, in the large-$N_c$ limit; and (ii) that the ratio of the width over the mass goes to a constant also in the same asymptotic limit, *i.e.*,

$$
\frac{\Gamma}{M(n)} = \frac{a}{N_c}\left(1 + \mathcal{O}\left(\frac{1}{N_c}, \frac{1}{n}\right)\right)\,, \qquad (n \gg 1)\,.
\tag{12}
$$

The scale $\Lambda_{QCD}$ is related to the string tension and is expected to be of order 1 GeV (see also below). The scale $F_0$ sets the normalization of the two-point function.

Both assumptions are supported by the solution of two-dimensional QCD [21], the string picture of hadrons [22] and phenomenology [23]. The picture that emerges is the following.

Starting from the dispersive representation obeyed by the Adler function, it is convenient to express it as a Borel–Laplace transform

$$
\begin{aligned}
\mathcal{A}(q^2) = -q^2 \frac{d\Pi(q^2)}{dq^2} &= -q^2 \int_0^\infty dt\, \rho(t) \int_0^\infty d\sigma\, \sigma\, \mathrm{e}^{-\sigma(t-q^2)} \\
&= -q^2 \int_0^\infty d\sigma\, \mathrm{e}^{\sigma q^2}\, \sigma \mathcal{B}^{[\rho]}(\sigma)\,,
\end{aligned}
\tag{13}
$$

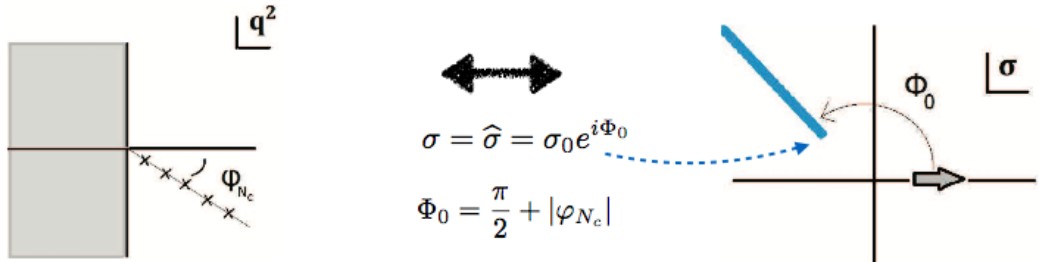

Figure 4: *Schematic representation of the connection between the singularities in the $q^2$ and $\sigma$ complex planes. The thick gray arrow in the right panel depicts the initial path taken in the sigma integral in Eq. (13).*

where

$$\mathcal{B}^{[\rho]}(\sigma) = \int_0^\infty dt\; \rho(t)\; e^{-\sigma t} \tag{14}$$

is the Laplace transform of the spectral function. The OPE corresponds to an expansion of $\mathcal{B}^{[\rho]}(\sigma)$ around $\sigma = 0$. We see that $\mathcal{B}^{[\rho]}(\sigma)$ is well-defined for $\mathrm{Re}\,\sigma > 0$, since $\rho(t)$ (*i.e.*, the spectral function) must go to a constant as $t \to \infty$, for finite $N_c$. Any singularities of $\mathcal{B}^{[\rho]}(\sigma)$ thus have to reside in the half-plane $\mathrm{Re}\,\sigma \le 0$. This representation of the Adler function in terms of $\mathcal{B}^{[\rho]}(\sigma)$ is valid for $\mathrm{Re}(\sigma q^2) < 0$, and for $\sigma > 0$ this means $q^2 < 0$. This is the key point: as one rotates $\sigma$ in the complex plane from $\mathrm{Re}\,\sigma > 0$ to $\mathrm{Re}\,\sigma < 0$, one is analytically continuing in the $q^2$ complex plane from $q^2 < 0$ to $q^2 > 0$. This is what we want.

If the spectrum $\rho(t)$ were to vanish for $t > t_0$, the function $\mathcal{B}^{[\rho]}(\sigma)$ would be analytic in the whole complex plane, and the above rotation in $\sigma$ would produce an OPE convergent for $|q^2| > t_0$. Of course, the spectrum goes all the way to infinity, as Eq. (11) clearly shows. The existence of an infinite number of poles of $\Pi(q^2)$ on the Minkowski axis for $N_c = \infty$ produces singularities for $\mathcal{B}^{[\rho]}(\sigma)$ on the imaginary axis in the $\sigma$ plane which, for the spectrum in Eq. (11), are branch points [8]. As $N_c$ evolves from infinity down to 3, the location of these poles recedes into the next Riemann sheet an angle given by

$$\varphi_{N_c} = -\frac{\Gamma}{M(n)} = -\frac{a}{N_c}\left(1 + \mathcal{O}\left(\frac{1}{N_c}, \frac{1}{n}\right)\right), \tag{15}$$

turning what were poles on the real $q^2 > 0$ axis into resonance peaks. Since $\sigma$ and $q^2$ are locked together, through Eq. (13), to satisfy $\mathrm{Re}(\sigma q^2) < 0$, this forces all branch points in the $\sigma$ plane off the imaginary axis, with the one closest to the origin moving to a position given by

$$\sigma = \hat{\sigma} \approx \frac{2\pi}{\Lambda_{QCD}^2}\, e^{i\Phi_0},$$
$$\Phi_0 \approx \frac{\pi}{2} + |\varphi_{N_c}|. \tag{16}$$

The position of the branch point is signaled by a blue arrow in Fig. 3.

In this situation, as the path in the $\sigma$ plane is rotated in the integral (13) from $\arg \sigma = 0$ to $\arg \sigma = \pi$, one sweeps through the blue line in Fig. 3, picking up a contribution given by

$$\text{Im}\,\Pi_{DV}(q^2) \sim e^{-2\pi \frac{a}{N_c}\frac{q^2}{\Lambda_{QCD}^2}} \sin\left[\frac{2\pi}{\Lambda_{QCD}^2}\left(q^2 - c - b\log\frac{q^2}{\Lambda_{QCD}^2}\right)\right]\left(1 + \mathcal{O}\left(\frac{1}{N_c}, \frac{1}{q^2}, \frac{1}{\log q^2}\right)\right) . \tag{17}$$

This expression can be parametrized as in Eq. (6), up to a small logarithmic corrections (since, for large $q^2$, $q^2 \gg b\log q^2$). In fact, QCD Regge phenomenology is consistent with this term $b$ being absent.

Besides the branch point (16), in principle there may be other branch points located in the same quadrant further away from the origin but, since the exponent in Eq. (17) is governed by the radial distance of these points to the origin, their contribution to $\text{Im}\,\Pi_{DV}$ will correspondingly contain a stronger exponential suppression. In this way, the expansion at large $q^2$ of $\text{Im}\,\Pi_{DV}$ becomes a combined series in $1/q^2$ and exponentials $e^{-q^2}$, of decreasing importance, as in the Theory of Hyperasymptotics [25].

Equation (17) connects the parameters from the radial Regge trajectories (11) to the parameters $\alpha_V$, $\beta_V$, $\gamma_V$ and $\delta_V$ of Eq. (6), which were obtained from fits involving the vector spectral function in $\tau$ decay. On the other hand, fits to meson spectroscopy give [23][3]

$$\Lambda_{QCD}^2 = 1.35(4) \text{ GeV}^2 , \quad \frac{\Gamma}{M} = 0.12(8) , \tag{18}$$

which translate into

$$\beta_V = \frac{2\pi}{\Lambda_{QCD}^2} = 4.7(2) \text{ GeV}^2 , \quad \gamma_V = \frac{2\pi}{\Lambda_{QCD}^2}\frac{a}{N_c} = 0.6(4) \text{ GeV}^{-2} . \tag{19}$$

These numbers are to be compared to the results from the fit involving $\tau$ data [17]:

$$\beta_V = 4.2(5) \text{ GeV}^{-2} \quad , \quad \gamma_V = 0.7(3) \text{ GeV}^{-2} . \tag{20}$$

The agreement is rather satisfactory. Notice in particular the importance of having the factors of $2\pi$ in Eq. (19).

## 4    Conclusion

We have argued that the mass of the $\tau$ lepton is not high enough to be able to dismiss the DV term (5) in the FESR (3) and that, because of that, one has to use a parametrization of the DV term which is physically sound, such as that given in Eq. (6). Attempts to work only at $s_0 = m_\tau^2$, assuming integrated DVs are negligible at this $s_0$ for doubly and triply pinched weights, run into the problem that the number of OPE parameters to be fit exceeds the number of spectral integrals available as input, unless, as in the truncated OPE strategy, one neglects sufficiently many higher-$D$ OPE contributions present in the analysis. We tested the reliability of the truncated OPE strategy, which neglects such higher-$D$ contributions, using EM FESRs employing recent $R(s)$ data as input, and found that this strategy, and the assumptions underlying it, fail badly. This leads us to the conclusion that one must

---

[3]For example, in the case of the $\rho$, one finds $\Gamma/M \simeq 0.19$.

take advantage of the $s_0$ dependence of $\tau$-based spectral integrals to have enough input to fit all relevant OPE parameters which, in turn, forces us to work at lower scales, where it becomes more important to take DVs into account. We conclude that an accurate extraction of $\alpha_s$ using $\tau$-decay data not subject to uncontrolled systematic errors requires a reasonable description of the DVs.

These conclusions stand in sharp contrast to the claims of Ref. [1]. The authors of Ref. [1] claim that DVs are sufficiently suppressed in the $ud\ V + A$ two-point function to be able to neglect them altogether when using doubly or triply pinched weights, and when working at the highest available scale, $s_0 = m_\tau^2$. The use of such weights, with their higher degrees, however, forces the authors of Ref. [1] to make strong assumptions about the behavior of the (asymptotic) OPE series, in particular, that contributions from higher-$D$ condensates present in the FESRs they employ can be neglected. These assumptions have been tested using the analogous EM FESRs and found to fail badly. This rules out the truncated OPE strategy employed in Ref. [1] as a reliable method for use in the hadronic $\tau$-decay determination of $\alpha_s$.

## Acknowledgements

MG and SP gratefully acknowledge interesting discussions with R. Miravitllas on the mathematical theory of Hyperasymptotics and its connection to Duality Violations. DB, KM and SP would like to thank the Department of Physics and Astronomy at San Francisco State University for hospitality. The work of D.B. is supported by the São Paulo Research Foundation (Fapesp) grant No. 2015/20689-9 and by the Brazilian National Council for Scientific and Technological Development (CNPq), grant No. 305431/2015-3. This material is based upon work supported by the U.S. Department of Energy, Office of Science, Office of High Energy Physics, under Award Number DE-SC0013682 (MG). KM is supported by a grant from the Natural Sciences and Engineering Research Council of Canada, and SP by CICYTFEDER-FPA2014-55613-P, 2014-SGR-1450 and the CERCA Program/Generalitat de Catalunya.

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
