# Peer review of "Determining $α_s$ from hadronic $τ$ decay: the pitfalls of truncating the OPE"

_SciPost Physics Proceedings_

## Round 1 · Referee Report · Anonymous (Referee 1) · 2018-12-9

Report

This contribution aims at showing that duality violations cannot be neglected in the determination of the QCD coupling from hadronic $\tau$ decays. While the analysis of duality violations is certainly an important topic that needs to be addressed case by case, I have some doubts about the adopted strategy and I miss some crucial information in the text. Therefore, I would like to ask the authors to consider the points described under "Requested changes", and modify the paper accordingly.

Requested changes

1. I believe that the authors do not show compelling evidence that the analysis in section 2 of $e^+e^-$ data (V channel) should imply the failure of OPE-based analyses applied to $\tau$-decay in the V+A channel. As I said, these issues must be addressed in detail case by case. The authors instead do not go beyond general claims, and these general claims are repeated many times in the paper. As the authors themselves observe, the V case is different from the V+A case. I would like the authors to limit their comments to those based on quantifiable and clearly formulated evidence.

2. The title is misleading, because it is an analysis of $e^+e^-$ data, see point 1. Maybe the authors can find a title that better captures the full content of the contribution.

3. I cannot find a clear error analysis of the theoretical predictions, e.g. the uncertainties associated to the curves shown in Figure 3, and how they could actually impact the $\tau$-decay determinations of $\alpha_s$. I would like the authors to address more precisely the theoretical uncertainties in their analysis, also by means of references to the appropriate published works.

4. The authors define a truncated-OPE strategy in section 2, where they assume that certain higher dimensional condensates can be neglected. They also seem to imply that this is what is done in other analyses which they criticise. I disagree. My understanding of the latter analyses is that the sensitivity of the result, e.g., $\alpha_s$, to higher dimensional condensates has been estimated through fits to data with varying weights. The same should be done for the example in Figure 3. I would like the authors to better formulate the related text.

5. In section 3 the authors discuss a model for duality violations. The parameters of this model are determined in specific ranges of $q^2$ and for specific processes. The authors then observe that the agreement of (20) with the results from the fit involving $\tau$ data is rather satisfactory. However, I could not find a complete analysis by the authors showing how their fitted parameterisation (20) behaves outside the fitted regions and for $\tau$-data versus $e^+e^-$ data. The authors should be more clear about this aspect and provide the relevant information.

  • validity: -
  • significance: -
  • originality: -
  • clarity: -
  • formatting: -
  • grammar: -

Author:  Santiago Peris  on 2018-12-13  [id 381]

(in reply to Report 1 on 2018-12-09)

Please find attached file (rebuttal.pdf).

Attachment:

rebuttal.pdf

Anonymous on 2019-01-17  [id 407]

(in reply to Santiago Peris on 2018-12-13 [id 381])

Dear Editor,

Please find attached our response to the second referee report.
Thank you.

Attachment:

Final-Rebuttal-2.pdf

Anonymous on 2019-01-08  [id 397]

(in reply to Santiago Peris on 2018-12-13 [id 381])

Contrary to what the authors claim, my comments are motivated by a careful reading of their contribution, whose content is in parts confused and in parts lacking necessary information.

Motivated by the answer of the authors, I add a few comments and related requests to the authors:

  1. According to the authors, Figure 3 of their contribution demonstrates the failure of the “truncated OPE” strategy employed in the EM FESR based on e+e- data with s0 ≥ mт2. The authors then claim that, as a consequence, Figure 3 also demonstrates the failure of the truncated OPE strategy used in Refs. [1,2] to determine αs with V+A т-decay data, thus, I insist, different data in a different channel — the authors show to be aware of the possible presence of corrections to the isospin limit and SU(3)F breaking effects, but they do not further discuss them —.

Moreover, Figure 3 shows effects on the EM spectral integrals whose size remains well below 0.0002 for s0 ≥ 3 GeV2, but nowhere else the authors explain how this numerical effect would quantitatively translate into a “failure” of the determination of αs in Refs. [1,2], or a failure of an analogous strategy applied to e+e- data.

When asked for the appropriate details, the authors replied that those details are part of a future publication and refused to provide them.

Why should I then accept for publication a contribution whose main aim is to criticise published work, i.e. Refs. [1,2], and whose criticism appears to be based on a claim rather than a complete and convincing analysis?

The authors are thus requested to provide the necessary quantitative information in a more clear discussion of their results. If they cannot provide further information in order to substantiate their claims, then they should remove the repeated criticism to other works.

  1. The second part of this contribution is again a criticism to Ref. [1]. The authors observe that duality violations (DV) are not negligible, also in V+A т-decay data. However, this contribution does not contain any quantitative argument on DV that could support their criticism.

The authors should give more space to quantitative evidence that directly applies to the example at hand, rather than criticising other works on too general or indirect grounds.

  1. In their answer the authors write: “As to the comparison of the corresponding DV oscillations between tau data and e+e- data, we refer the referee to Fig. 5 of Ref. [20].” I have looked at Fig. 5 of Ref. [20] and my conclusion is that the orange curve could well be overestimating DV effects when compared to the grey, blue and red data. In my opinion this may cast some doubts on the estimate of DV effects in т-data and I cannot agree with the conclusions of the authors.

As the authors can see, I am not quite agreeing with their conclusions on the effects of DV. Therefore, I ask the authors to better explain and substantiate their conclusions. On the other hand, the authors may also choose to expand the first part of this contribution on e+e- and leave out the second part.

  1. Eq. (20) of section 3 provides the values of two of the four parameters of the DV model proposed by the authors and determined by a fit of т-data. Again, this is an example of incomplete information. What is the use of these two numerical values, if no information is provided on the other two parameters, their associated uncertainties and the s dependence of the fit results?

The authors should provide complete information, otherwise this last part of the contribution is useless.

In conclusion, the authors should address the four points above and make the necessary modifications before this contribution can be considered for publication.

---

## Editorial Decision

resubmitted